# Geometric Pocket-Centric Protein Encoding for Polypharmacology-Guided Multi-Target Drug Design

Haoran Liu [1]   Xiaoli Lin [1]   Jing Hu [1]   Yu Zou [2]   Xiaolong Zhang [1]

## Abstract

Polypharmacology provides a powerful strategy for treating complex diseases, but identifying molecules that simultaneously satisfy coupled constraints across multiple biological targets remains difficult. Existing methods typically model protein pockets in isolation and struggle to jointly account for multiple heterogeneous binding sites when designing a single shared ligand. To address these limitations, we propose a pocket-structure-centric generative framework for polypharmacology. This framework introduces a novel protein topological representation that selectively masks ligand-irrelevant residues while explicitly modeling backbone folding geometry and inter-residue spatial proximity within binding pockets. In addition, structural representations are jointly fused with amino acid and nucleotide sequences to capture their complementary information across targets. Experiments on COVID-19, schizophrenia, and tumor targets show that this framework generates valid candidates with significantly improved binding affinities compared to state-of-the-art methods.

## 1. Introduction

Machine learning has demonstrated substantial potential in accelerating drug discovery (Wong et al., 2024). Complex diseases, such as cancer, often arise from intricate and interdependent biological pathways (Munson et al., 2024). Drugs that exert therapeutic effects by modulating the activity of specific target proteins are commonly referred to as targeted therapies (Liu et al., 2024). To date, most computational drug discovery studies have focused on single-target drug design (Liu et al., 2025). However, single-target drugs are prone to resistance and may inadvertently activate compensatory signaling pathways (Wu et al., 2026), reducing efficacy or causing adverse effects (Yang et al., 2024).

As a promising alternative, dual-target drugs have gained increasing attention (Layman et al., 2024), supported by the growing number of FDA-approved multi-target therapeutics (Lin et al., 2024). By simultaneously interacting with multiple disease-relevant targets (Lin et al., 2025), dual-target drugs can achieve enhanced therapeutic efficacy and robustness against pathway redundancy (Lu et al., 2025). Nevertheless, discovering such compounds remains highly challenging (Wan et al., 2025). The chemical space is vast-estimated at approximately $10^{60}$ molecules-and multi-target drug discovery requires candidates to exhibit favorable affinity toward multiple targets simultaneously (Reymond et al., 2012), imposing coupled constraints on medicinal chemistry properties and biological activity (Ye et al., 2022).

Recent advances in deep generative modeling have enabled ligand-based strategies for multi-target drug discovery (Munson et al., 2024). Munson propose POLYGON, which formulates polypharmacological drug design as a generative reinforcement learning problem. POLYGON represents molecules using SMILES strings and generates dual-target candidates optimized for predicted bioactivity. Despite its effectiveness, POLYGON adopts a purely ligand-centric perspective and does not explicitly model three-dimensional molecular structures. As a result, it is limited in capturing structure-driven binding mechanisms and may struggle to generalize across targets with diverse and heterogeneous structural constraints.

Subsequent work incorporates explicit three-dimensional molecular representations (Yuan et al., 2025). Yuan et al. propose MDRL, which combines diffusion models and reinforcement learning to generate polypharmacological compounds directly in 3D space. This design captures spatial chemical information beyond SMILES and jointly optimizes properties such as QED and synthetic accessibility. However, MDRL remains molecule-centric: target proteins are

[1] Hubei Key Laboratory of Intelligent Information Processing and Real-Time Industrial System, School of Computer Science and Technology, Wuhan University of Science and Technology, Wuhan, China [2] Hubei Key Laboratory of Occupational Hazard Identification and Control, School of Medicine, Wuhan University of Science and Technology, Wuhan, China. Correspondence to: Xiaoli Lin <linxiaoli@wust.edu.cn>, Yu Zou <zouyu@wust.edu.cn>, Xiaolong Zhang <xiaolong.zhang@wust.edu.cn>.

*Proceedings of the 43rd International Conference on Machine Learning*, Seoul, South Korea. PMLR 306, 2026. Copyright 2026 by the author(s).

only implicitly modeled via activity predictors or docking-based rewards, and protein structures do not guide the initial generation process.

Conditional generative models further incorporate target-related information (Zhou et al., 2025). Zhou propose an E(3)-equivariant conditional generative framework for dual-target drug design, where docking-predicted binding affinities are embedded as conditional vectors to guide 3D molecular generation. Although protein information is leveraged through affinity-based embeddings, protein structures are only indirectly involved and compounds are primarily filtered post hoc. Consequently, structural information influences selection rather than generation, leading to computationally intensive pipelines with limited scalability.

Expert-guided rational design has also been explored (Lu et al., 2025). Lu demonstrate that dual-target agonists can be rationally designed by exploiting structural similarities between binding pockets and chemical similarities among known agonists. This paradigm yields highly potent and specific dual-target compounds with favorable pharmacological profiles. However, it relies heavily on strong prior knowledge of pocket similarity across targets, which limits its generalization to unseen targets and its applicability to large-scale or automated drug discovery.

In summary, existing methods either omit protein structural information during molecular generation or incorporate it only through post hoc evaluation and expert priors. In contrast, our approach integrates protein structure directly into the design stage via a pocket-centric representation that jointly encodes 3D binding geometry, amino acid sequences, and nucleotide information. By enforcing multiple pharmacological properties as explicit constraints during generation, our model directly reasons about multi-target protein-ligand interactions while producing compounds that satisfy desired medicinal chemistry profiles, thereby improving both efficiency and generalization.

We summarize our contributions as follows:

- We propose a novel protein structure representation tailored for multi-target drug design. The representation models protein topology through folding angles and spatial distances, while suppressing ligand-irrelevant internal residues to refine the characterization of potential binding pockets.

- We further introduce a nonlinear graph attention-based protein structure encoder that encodes pocket-aware structural interactions at the amino-acid level, enabling the aggregation of ligand-shareable binding patterns across multiple target proteins.

- We present a molecule generation framework guided by multimodal target features and physicochemical

constraints. The generator integrates protein structures, amino acid and nucleotide sequences, and eight drug-like properties to produce candidate molecules optimized for multiple targets simultaneously.

- We validate the proposed multi-target drug design framework across diverse disease-related target combinations. Our method is capable of generating both inhibitors and agonists, achieving strong binding affinities and consistently outperforming recent state-of-the-art approaches.

## 2. Preliminary

We consider multi-target drug design over a heterogeneous set $\mathcal{T}$ of $K$ biological targets. Each target $\mathcal{P}_k \in \mathcal{T}$ consists of a three-dimensional structural topology $\mathcal{G}_k$, an amino acid sequence $\mathbf{S}_{aa,k}$, and a nucleotide sequence $\mathbf{S}_{nt,k}$. We formulate multi-target drug design as a conditional molecular generation problem:

$$P(\mathbf{c} \mid \mathcal{T}, \mathbf{\Phi}), \qquad (1)$$

where $\mathbf{c}$ denotes the candidate compound to be generated and $\mathbf{\Phi} \in \mathbb{R}^8$ represents a vector of desired pharmacological properties.

**Kolmogorov Arnold Networks (KAN):** Unlike Multi-Layer Perceptrons (MLPs) with fixed activation functions, KANs model high-dimensional nonlinear mappings using learnable univariate functions parameterized by B-spline bases:

$$\text{KAN}(\mathbf{x}) = \sum_{j=1}^{n_{\text{in}}} \psi_j(x_j), \qquad (2)$$

$$\psi_j(x_j) = \sum_i w_{j,i} \, \text{spline}_i(x_j), \qquad (3)$$

where $\mathbf{x} \in \mathbb{R}^{n_{\text{in}}}$ is the input vector. Each $\psi_j(\cdot)$ is a learnable univariate nonlinear function, where $\text{spline}_i(\cdot)$ denotes the $i$-th B-spline basis function and $w_{j,i}$ is its corresponding learnable coefficient. This formulation enables flexible piecewise-smooth nonlinearities, making KANs well suited for modeling non-monotonic biological features.

## 3. Proposed Approach

We propose a multi-target drug design framework that integrates protein structures, amino acid sequences, nucleotide sequences, and explicit pharmacological property constraints into a unified conditional generation paradigm as shown in Figure 1. The framework models protein binding landscapes and generates compounds compatible with multiple targets under explicit pharmacological constraints.

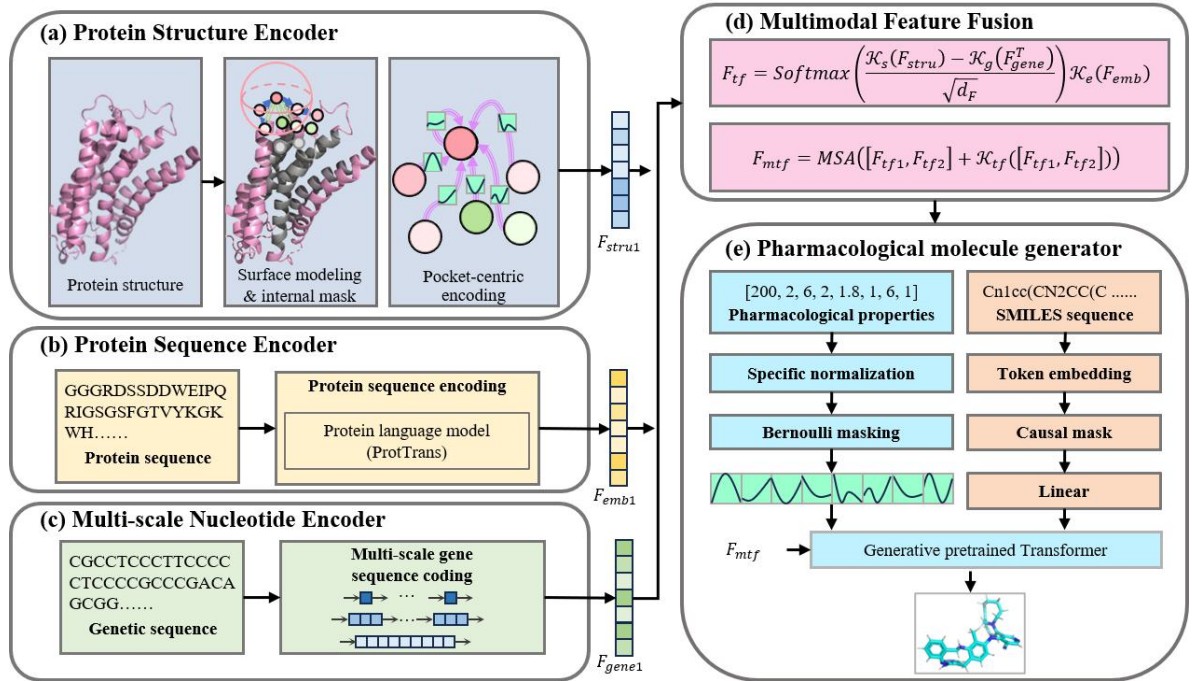

*Figure 1.* Overview of the proposed multi-target drug design framework. (a) Pocket-aware protein structure encoder of binding landscapes. (b) Protein sequence encoder. (c) Multi-scale nucleotide sequence encoder. (d) Multimodal feature fusion. (e) Pharmacological molecule generator guided by fused multi-target features and pharmacological constraints.

## 3.1. Protein Structure Encoder

As shown in Figure 2, we develop a geometric-topological representation that jointly models backbone folding and pocket-scale spatial proximity while masking residues irrelevant to ligand binding. Building on this representation, we further introduce a pocket-centric KAN-GAT encoder that selectively propagates information within coherent pocket regions, enabling the capture of heterogeneous yet compatible binding pockets across targets.

### 3.1.1. GEOMETRIC-TOPOLOGICAL MODELING OF PROTEIN BINDING LANDSCAPES

Drug-protein interactions rely on precise geometric complementarity, which is inadequately captured by conventional residue-level graphs. We propose a protein topological representation that jointly models peptide backbone folding and inter-residue spatial proximity, enabling characterization of binding-pocket curvature critical to ligand design.

Standard representations encode residue relations using $C_\alpha$-$C_\alpha$ Euclidean distances, which can distort local geometry: although sequential residues are connected by short C-N peptide bonds ($< 1.5$ Å), their $C_\alpha$-$C_\alpha$ distances typically exceed 3 Å. To address this limitation, our representation integrates backbone-induced conformational relations for adjacent residues with non-local geometric coupling among spatially proximal residues within the binding pocket. Residues

unlikely to participate in ligand binding are masked, while spatially distant residues remain disconnected.

We model the structural information of each target protein $\mathcal{P}_k$ as a topological representation $\mathcal{G}_k = (\mathcal{V}_k, \mathcal{E}_k)$, where each node $v_i \in \mathcal{V}_k$ corresponds to an amino acid residue and is associated with the three-dimensional coordinates of its $C_\alpha$ atom, $\mathbf{p}_i \in \mathbb{R}^3$. Edges in $\mathcal{E}_k$ encode both spatial proximity within the binding pocket and local backbone folding geometry.

Consider a target protein with an amino acid sequence A, B, C, and D, where the $C_\alpha$ atoms of residues A, B, C, and D are located at coordinates $\mathbf{p}_1, \mathbf{p}_2, \mathbf{p}_3$, and $\mathbf{p}_4 \in \mathbb{R}^3$, respectively.

**Pocket-scale Non-local Geometric Relations:** To capture pocket-scale residue interactions beyond sequence adjacency, we introduce non-local geometric coupling relations between residues that are spatially proximal within the binding landscape. For two non-adjacent residues A and D that fall within the binding-pocket scale, we define a spatial edge:

$$e_{AD}^{(s)} = \frac{1}{\|\mathbf{p}_1 - \mathbf{p}_4\|_2^2}, \tag{4}$$

where $\|\cdot\|_2^2$ denotes the squared Euclidean norm. This formulation assigns larger weights to spatially closer residues and captures distance-dependent interactions that are critical for ligand binding.

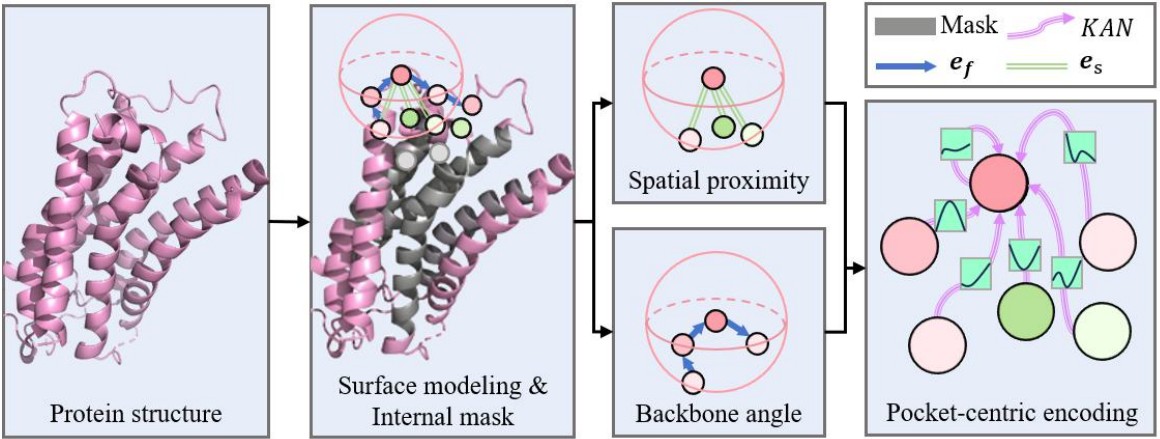

*Figure 2.* Pocket-centric protein structure encoder.

**Backbone-induced Conformational Relations:** To preserve local folding geometry along the peptide backbone, we encode backbone-induced conformational relations between sequentially adjacent residues. For two sequentially adjacent residues B and C along the peptide backbone, we define a folding edge based on the angle between consecutive backbone segments:

$$e_{BC}^{(f)} = \arccos\left(\frac{(\mathbf{p}_2 - \mathbf{p}_1) \cdot (\mathbf{p}_3 - \mathbf{p}_2)}{\|\mathbf{p}_2 - \mathbf{p}_1\|_2 \, \|\mathbf{p}_3 - \mathbf{p}_2\|_2}\right). \quad (5)$$

Together, non-local geometric coupling and backbone-induced conformational relations provide a dual-view characterization of protein binding landscapes, capturing pocket-scale residue proximity and local backbone geometry. By jointly incorporating spatial edges $e^{(s)}$ and folding edges $e^{(f)}$ into $\mathcal{G}_k$, this representation yields a compact yet expressive description of binding pockets beyond distance-based graphs, while preserving features critical for drug-target interactions.

### 3.1.2. POCKET-CENTRIC ENCODING OVER PROTEIN TOPOLOGIES

Building upon the dual-view protein topology, we design a pocket-centric structural encoder that captures spatial and folding interactions using lightweight attention while remaining computationally efficient. Message passing is restricted to spatially coherent pocket regions by pruning long-range residue interactions beyond the pocket scale, and all node-wise linear projections are replaced with nonlinear KAN transformations to enhance expressive power under limited parameter budgets.

We denote the feature of node $i$ as $v_i \in \mathbb{R}^d$, the edge attribute between nodes $i$ and $j$ as $e_{ij}$, and the neighbor set of node $i$ as $\mathcal{N}(i)$. KAN-based nonlinear operators $\mathcal{K}_\ell$ are used for node transformations, with $\sigma_n(\cdot)$ for layer normaliza-

tion, $\phi(\cdot)$ for nonlinear activation, and $\oplus$ indicating residual connections.

The first layer updates nodes as:

$$h_i^{(1)} = \sigma_n\left(v_i \oplus \phi\left(\sum_{j \in \mathcal{N}(i)} \alpha_{ij}^{(1)} \mathcal{K}_1(v_j)\right)\right), \quad (6)$$

with attention coefficients:

$$\alpha_{ij}^{(1)} = \sigma_a\left((\mathcal{K}_1(v_i) \cdot \mathcal{K}_1(v_j))/\sqrt{d} + \sum_k e_{ij,k}/\sqrt{d_e}\right), \quad (7)$$

where $d$ is the node feature dimension and $d_e$ the edge feature dimension. $\sigma_a(\cdot)$ denotes the attention normalization function.

The second layer refines node representations similarly:

$$h_i^{(2)} = \sigma_n\left(h_i^{(1)} \oplus \phi\left(\sum_{j \in \mathcal{N}(i)} \alpha_{ij}^{(2)} \mathcal{K}_2(h_j^{(1)})\right)\right), \quad (8)$$

with attention coefficients defined as:

$$\alpha_{ij}^{(2)} = \sigma_a\left((\mathcal{K}_2(h_i^{(1)}) \cdot \mathcal{K}_2(h_j^{(1)}))/\sqrt{d} + \sum_k e_{ij,k}/\sqrt{d_e}\right). \quad (9)$$

Node embeddings are aggregated via mean and max pooling:

$$x_{\text{graph}} = \text{Concat}\left(\text{MeanPool}(\{h_i^{(2)}\}), \text{MaxPool}(\{h_i^{(2)}\})\right), \quad (10)$$

followed by KAN-based pooling and a fully connected layer with layer normalization:

$$F_{\text{stru}} = \sigma_n\left(W \mathcal{K}_{\text{pool}}(x_{\text{graph}}) + b\right), \quad (11)$$

producing a fixed-dimensional embedding $F_{\text{stru}} \in \mathbb{R}^{1024}$.

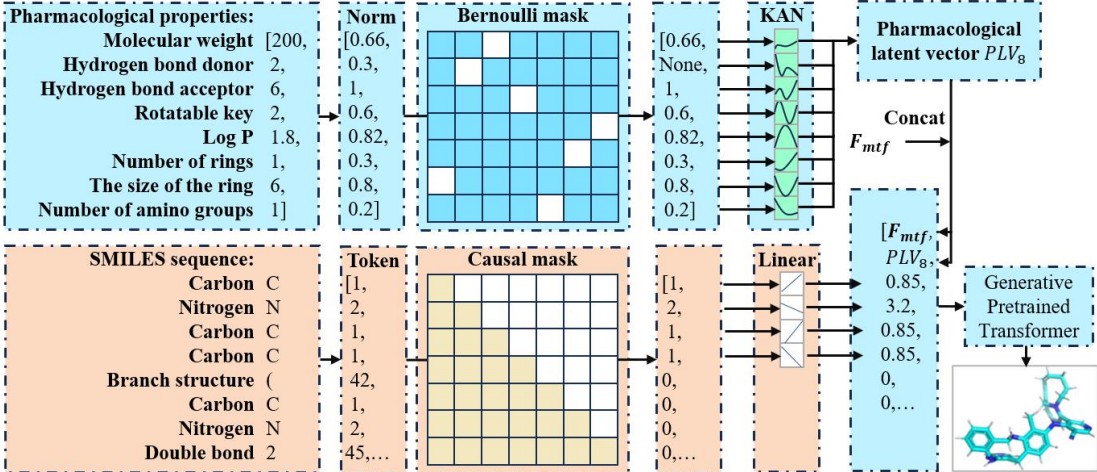

*Figure 3.* Pharmacological molecule generator.

By integrating KAN-based nonlinear projections with attention-driven message passing, the proposed encoder effectively captures fine-grained pocket geometry while maintaining translation and rotation invariance. This design highlights druggable regions across heterogeneous targets and produces robust structural representations for downstream multi-target molecular generation.

### 3.2. Protein Sequence Encoder

The sequence $S_{aa}$ serves as a complement to the 3D structure, especially for disordered regions where crystal data is unavailable. We employ a pre-trained ProtTrans (Elnaggar et al., 2021) model as the sequence encoder:

$$F_{emb} = \text{ProtTrans}(S_{aa}). \qquad (12)$$

The attention pooling mechanism ensures that the model ignores generic scaffold sequences and focuses on conserved functional domains.

### 3.3. Multi-scale Nucleotide Encoder

Genomic sequences $S_{nt}$ contain latent regulatory signals. We employ a multi-scale CNN to capture patterns at three genomic resolutions:

$$f_k = \text{ReLU}\big(\text{Norm}(\text{Conv1D}_k(S_{nt}))\big), \qquad (13)$$

$$F_{gene} = \text{GlobalPool}\big(\text{Concat}(f_1, f_3, f_9)\big). \qquad (14)$$

Here, $f_k$ is the feature map from a one-dimensional convolution with kernel size $k \in \{1, 3, 9\}$, and $F_{gene}$ denotes the pooled nucleotide representation. The scales $\{1, 3, 9\}$ capture base level, codon level, and neighboring level patterns, respectively.

### 3.4. Multimodal Feature Fusion

Modality-specific embeddings ($F_{stru}$, $F_{emb}$, $F_{gene} \in \mathbb{R}^{1024}$) are fused within each target and then aggregated across targets. High-order nonlinear transformations are implemented using KAN-based operators to enhance cross-modal interactions. The three modalities are fused into a target-level representation via attention:

$$F_{tf} = \text{SoftMax}\left(\frac{\mathcal{K}_s(F_{stru}) \cdot \mathcal{K}_g(F_{gene})^\top}{\sqrt{d_F}}\right) \mathcal{K}_e(F_{emb}). \qquad (15)$$

Target-level features are further aggregated across targets using residual multi-head self-attention:

$$F_{mtf} = \text{MSA}\big([F_{tf1}, F_{tf2}] + \mathcal{K}_{tf}([F_{tf1}, F_{tf2}])\big), \qquad (16)$$

yielding the final multi-target multimodal representation $F_{mtf}$.

### 3.5. Pharmacological Molecule Generator

Molecule generation is conditioned on eight pharmacological properties, as illustrated in Figure 3. These properties are selected to improve general drug-likeness by regulating molecular weight (MW), H-bond donors (HBD), H-bond acceptors (HBA), rotatable bonds (RotB), and the octanol-water partition coefficient (LogP) and to satisfy target-specific requirements. For instance, the TAAR1 receptor possesses a highly constrained binding pocket that favors ligands with ring sizes no larger than six atoms and requires agonist-induced activation. Accordingly, number of rings (NumR), maximum ring size (MaxR), and number of amino groups (NumA) are explicitly controlled. All properties are normalized to the unit interval $[0, 1]$ via z-score or min-max scaling, with detailed in Appendix A.

To improve robustness and generalization, a Bernoulli mask

**Algorithm 1** Pharmacology-guided Multi-Target Drug Design

---

1: **Input:** Targets $\mathcal{T} = \{\mathcal{P}_k\}_{k=1}^K$, $\mathcal{P}_k = (\mathcal{G}_k, \mathbf{S}_{aa,k}, \mathbf{S}_{nt,k})$, properties $\Phi$
2: **Output:** Molecule $\mathbf{c}$
3: **for** each $\mathcal{P}_k \in \mathcal{T}$ **do**
4: $\quad F_{tf,k} \leftarrow \mathrm{KAN}_{fuse}(\mathrm{Enc}_{stru}(\mathcal{G}_k) \oplus \mathrm{Enc}_{aa}(\mathbf{S}_{aa,k}) \oplus \mathrm{Enc}_{nt}(\mathbf{S}_{nt,k}))$
5: **end for**
6: $F_{mtf} \leftarrow \mathrm{MSA}(\{F_{tf,k}\}_{k=1}^K)$
7: $\hat{\Phi} \leftarrow \mathrm{Mask}(\Phi)$
8: $F_{cond} \leftarrow \mathcal{K}_{prop}(F_{mtf} \oplus \hat{\Phi})$
9: **while** not end **do**
10: $\quad \mathbf{h}_t \leftarrow \mathrm{Transformer}(\mathrm{Emb}(c_{<t}) + \mathrm{Proj}(F_{cond}))$
11: $\quad$ Sample $c_t$ and append to $\mathbf{c}$
12: **end while**
13: **Return c**

---

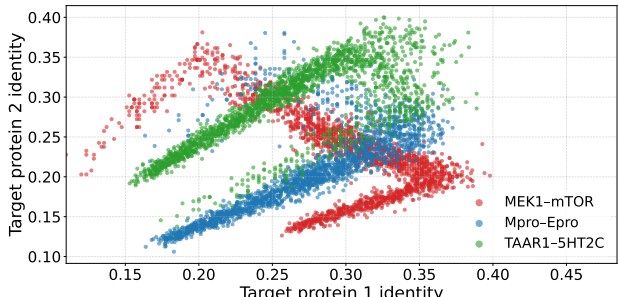

*Figure 4.* Low sequence homology between training and test proteins across COVID-19, SCZ, and Tumour.

is applied to the property vector to randomly mask a subset of attributes across training epochs:

$$\hat{\Phi} = \Phi \odot \mathbf{m} + \Phi_{null} \odot (1 - \mathbf{m}), \quad \mathbf{m} \sim \mathcal{B}(1 - p), \quad (17)$$

$\Phi_{null}$ is a learnable null embedding and $\mathbf{m}$ is a binary mask. This strategy enables the model to handle partially specified property constraints while mitigating overfitting to specific attribute combinations.

The conditioned latent vector is then obtained by fusing masked properties with the multi-target embedding via a KAN-based projection:

$$F_{cond} = \mathcal{K}_{prop}(F_{mtf} \oplus \hat{\Phi}). \quad (18)$$

This vector acts as a prefix bias in the Transformer:

$$\mathbf{h}_t = \mathrm{Transformer}\big(\mathrm{Emb}(s_{<t}) + \mathrm{Pos} + \mathrm{Proj}(F_{cond})\big). \quad (19)$$

The compound $\mathbf{c}$ is generated autoregressively as a molecular token sequence, where each hidden state $\mathbf{h}_t$ parameterizes the conditional distribution of the next token. Conditioning on $\mathcal{T}$ and $\Phi$ is injected via $F_{cond}$, ensuring that $P(\mathbf{c} \mid \mathcal{T}, \Phi)$ is globally guided by multi-target information and pharmacological constraints (see Algorithm 1).

## 4. Experiments

### 4.1. Dataset and Preprocessing

We construct a multimodal, multi-target dataset. Molecular compounds are collected from ZINC (Irwin et al., 2020) and ChEMBL (Zdrazil et al., 2024), retaining 90,432 ligands annotated with at least two targets. Protein sequences are obtained from UniProt, nucleotide sequences from NCBI, and predicted structures from AlphaFold (Varadi et al., 2022).

We estimate binding pocket extent using maximum interatomic distances, with 90% of molecules below 20.45 Å, and mask spatial edges beyond 20 Å. Details are provided in Appendix B.

Evaluation is conducted on three disease settings with experimentally resolved PDB structures: tumour (MEK1 (Gonzalez-Del Pino et al., 2021), mTOR (Choi et al., 1996)), COVID-19 (Mpro (Funk et al., 2024), Epro (Mandala et al., 2020)), and schizophrenia (TAAR1 (Shang et al., 2023), 5HT2C (Peng Y et al., 2018)). Training proteins sharing over 40% sequence homology with any test protein are removed to ensure target-level generalization (Figure 4). After filtering, the training dataset comprises 75,921 high-quality samples covering 1,305 proteins, all sourced from publicly available databases. Preprocessing details are in Appendix C. Disease-target mappings are provided in Appendix D.

### 4.2. Protein Topological Construction

We construct topology that capture key pocket and ligand-interacting features while reducing overall complexity (Appendix E). Internal residues irrelevant to binding are masked, and edges are selectively modeled using spatial distances and backbone folding angles. This preserves pharmacologically critical interactions while simplifying densely connected regions, enabling more efficient topology-based learning without losing essential structural information.

### 4.3. Pharmacology-based Molecular Generation

We perform principal component analysis (PCA) on uniformly sampled compounds across training epochs using standardized eight-dimensional pharmacological property vectors, with ranges reported in Appendix F (Table 5). A global two-dimensional PCA model is fitted, and compounds from epochs 1-4 are projected into the same space for comparison. For the COVID-19 and tumour tasks, all properties are sampled uniformly, whereas for the TAAR1-related schizophrenia task, the number of amino groups is

*Table 1.* MAE results across different property sets and epochs.

| Property | Epoch | MW | HBD | HBA | RotB | LogP | NumR | MaxR | NumA |
|----------|-------|------|------|------|------|------|------|------|------|
| Set 1 | 1 | 0.03 | 0.07 | 0.17 | 0.09 | 0.10 | 0.07 | 0.16 | 0.12 |
| Set 1 | 2 | 0.01 | 0.06 | 0.13 | 0.06 | 0.08 | 0.06 | 0.07 | 0.11 |
| Set 1 | 3 | 0.01 | 0.06 | 0.11 | 0.04 | 0.07 | 0.06 | 0.05 | 0.12 |
| Set 1 | 4 | 0.01 | 0.05 | 0.09 | 0.05 | 0.07 | 0.06 | 0.05 | 0.12 |
| Set 2 | 1 | 0.09 | 0.10 | 0.07 | 0.10 | 0.11 | 0.07 | 0.13 | 0.15 |
| Set 2 | 2 | 0.08 | 0.09 | 0.04 | 0.08 | 0.09 | 0.06 | 0.08 | 0.12 |
| Set 2 | 3 | 0.08 | 0.09 | 0.02 | 0.07 | 0.08 | 0.06 | 0.08 | 0.13 |
| Set 2 | 4 | 0.08 | 0.08 | 0.02 | 0.06 | 0.08 | 0.06 | 0.08 | 0.13 |
| Set 3 | 1 | 0.16 | 0.16 | 0.08 | 0.14 | 0.14 | 0.16 | 0.15 | 0.20 |
| Set 3 | 2 | 0.10 | 0.10 | 0.06 | 0.09 | 0.09 | 0.10 | 0.08 | 0.21 |
| Set 3 | 3 | 0.09 | 0.09 | 0.06 | 0.08 | 0.09 | 0.09 | 0.08 | 0.20 |
| Set 3 | 4 | 0.09 | 0.09 | 0.07 | 0.09 | 0.08 | 0.09 | 0.08 | 0.18 |
| Set 4 | 1 | 0.27 | 0.26 | 0.18 | 0.22 | 0.21 | 0.27 | 0.18 | 0.28 |
| Set 4 | 2 | 0.18 | 0.12 | 0.11 | 0.14 | 0.15 | 0.16 | 0.14 | 0.31 |
| Set 4 | 3 | 0.15 | 0.11 | 0.12 | 0.12 | 0.12 | 0.15 | 0.16 | 0.31 |
| Set 4 | 4 | 0.14 | 0.11 | 0.11 | 0.13 | 0.10 | 0.15 | 0.15 | 0.26 |
| Set 5 | 1 | 0.36 | 0.35 | 0.27 | 0.32 | 0.28 | 0.36 | 0.24 | 0.39 |
| Set 5 | 2 | 0.27 | 0.18 | 0.18 | 0.21 | 0.22 | 0.25 | 0.21 | 0.41 |
| Set 5 | 3 | 0.23 | 0.16 | 0.18 | 0.15 | 0.17 | 0.23 | 0.23 | 0.41 |
| Set 5 | 4 | 0.22 | 0.17 | 0.15 | 0.16 | 0.12 | 0.24 | 0.23 | 0.38 |

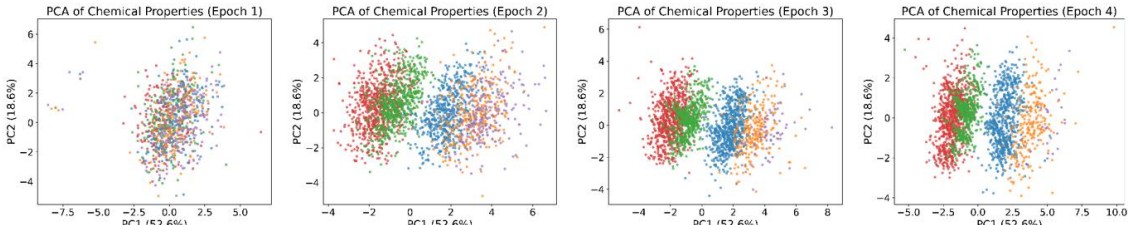

*Figure 5.* PCA of compound properties from epoch 1 to epoch 4.

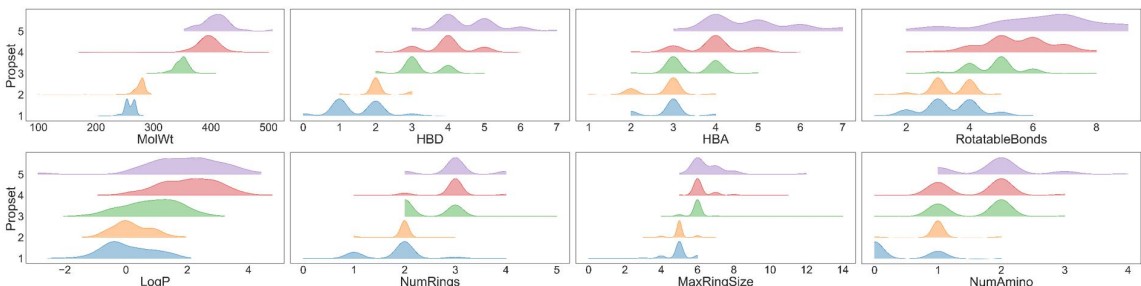

*Figure 6.* Ridge plot of attribute distribution of multiple compounds generated by Epoch4.

fixed to one and the maximum ring size is limited to six atoms, with remaining properties sampled uniformly.

As shown in Figure 5, generated molecules evolve from dispersed to structured, property-aligned distributions across epochs, reflecting progressively stronger conditioning on the target property vectors. Consistent with the Mean Absolute Error (MAE) results in Table 1, most attributes exhibit substantial optimization from Epoch 1 to 2, followed by

*Table 2.* Comparison of predicted binding affinities across different methods and targets.

| Method | Disease | Modal | Target | Binding Affinity |
|---|---|---|---|---|
| Munson's Method | Tumour | Ligand SMILES | MEK1 mTOR | -7.4 -8.4 |
| Yuan's Method | Tumour | Ligand Structure | MEK1 mTOR | -11.9 **−14.1** |
| Zhou's Method | COVID-19 | Binding Affinity | Mpro Epro | -10.3 -10.9 |
| Lu's Method | SCZ | Binding Pocket | TAAR1 5HT2C | -6.2 -6.8 |
| Ours | COVID-19 | Multi-modal | Mpro Epro | **−11.4** **−12.3** |
| | SCZ | Multi-modal | TAAR1 5HT2C | **−13.3** **−12.1** |
| | Tumour | Multi-modal | MEK1 mTOR | **−12.5** -11.6 |

finer refinement through Epoch 4. Molecules from Epoch 4 form more compact clusters with reduced variance, while the marginal property distributions in Figure 6 become increasingly concentrated within desired ranges, indicating improved stability and controllability. Notably, the first three property sets show stronger convergence at Epoch 4, likely due to the combined effects of dataset filtering, drug-like pretraining, and multi-target training data, whereas the latter two remain comparatively more dispersed.

### 4.4. Binding Affinity of Generated Molecule

We perform molecular docking simulations for each disease case against two protein targets. Binding affinities are computed as semi-empirical free energy changes, where lower values correspond to stronger binding. For clearer visualization, we report the absolute values of the Vina scores.

As shown in Figure 7, compounds are ranked along the x-axis in descending order according to their aggregate affinity, defined as the sum of binding affinities across both targets (black circles). Affinities for individual targets are overlaid as colored markers to illustrate their respective contributions. The top ten candidates are emphasized using enlarged markers and a shaded background, highlighting the model's ability to identify high-affinity lead compounds that simultaneously engage multiple biological pockets.

### 4.5. Comparison and Analysis

Table 2 summarizes the binding affinities and input modalities of different methods across disease settings. For COVID-19, the approach of Zhou (Zhou et al., 2025) achieves docking scores of −10.3 kcal/mol on Mpro and

−10.9 kcal/mol on Epro. For schizophrenia (SCZ), Lu (Lu et al., 2025) reports affinities of −6.2 kcal/mol on TAAR1 and −6.8 kcal/mol on 5HT2C. For tumour-related targets, Yuan (Yuan et al., 2025) attains −11.9 kcal/mol on MEK1 and −14.1 kcal/mol on mTOR, while Munson (Munson et al., 2024) reports comparatively weaker affinities of −8.4 kcal/mol on MEK1 and −7.4 kcal/mol on mTOR.

Our method consistently generates high-affinity multi-target molecules across all evaluated diseases. For COVID-19, it achieves binding affinities of −11.4 kcal/mol on Mpro and −12.3 kcal/mol on Epro. For SCZ, it reaches −13.3 kcal/mol on TAAR1 and −12.1 kcal/mol on 5HT2C, outperforming all baseline methods. On tumour targets, our approach attains −12.5 kcal/mol on MEK1 and −11.6 kcal/mol on mTOR, exceeding Munson (Munson et al., 2024) on both targets, and surpassing Yuan (Yuan et al., 2025) on MEK1 while remaining competitive on mTOR. Overall, these results demonstrate that our framework reliably produces multi-target molecules with strong and balanced binding affinities across diverse diseases.

These findings also highlight general limitations of existing multi-target design approaches, which often struggle to simultaneously balance affinities across multiple targets or efficiently incorporate protein structural information. By explicitly integrating multimodal protein representations with pharmacological constraints during generation, our framework overcomes these challenges, producing multi-target molecules with consistently strong and balanced binding across diverse disease settings.

### 4.6. Case Study

We analyze the binding conformations of a representative multi-target molecule against tumour-related targets, MEK1 (PDB ID: 7M0Y) and mTOR (PDB ID: 1FAP). As shown in Figure 8, the left panel depicts the binding pose in MEK1, while the right corresponds to mTOR. Protein structures are rendered as pink cartoons, binding pockets are highlighted in yellow, and ligands are shown in cyan. Key intermolecular interactions are annotated: green sticks indicate van der Waals contacts, green dashed sticks represent C-H bonds, dark purple denotes $\pi$-$\pi$ stacking, and light purple highlights alkyl or $\pi$-alkyl interactions. Despite the distinct pocket geometries of MEK1 and mTOR, the molecule adopts stable, target-specific binding conformations in both cases, demonstrating effective pocket-aware multi-target binding.

## 5. Conclusion

We propose a pocket-centric generative method for polypharmacology-guided multi-target drug design. The core of our method is a compact geometric-topological protein representation that integrates backbone folding ge-

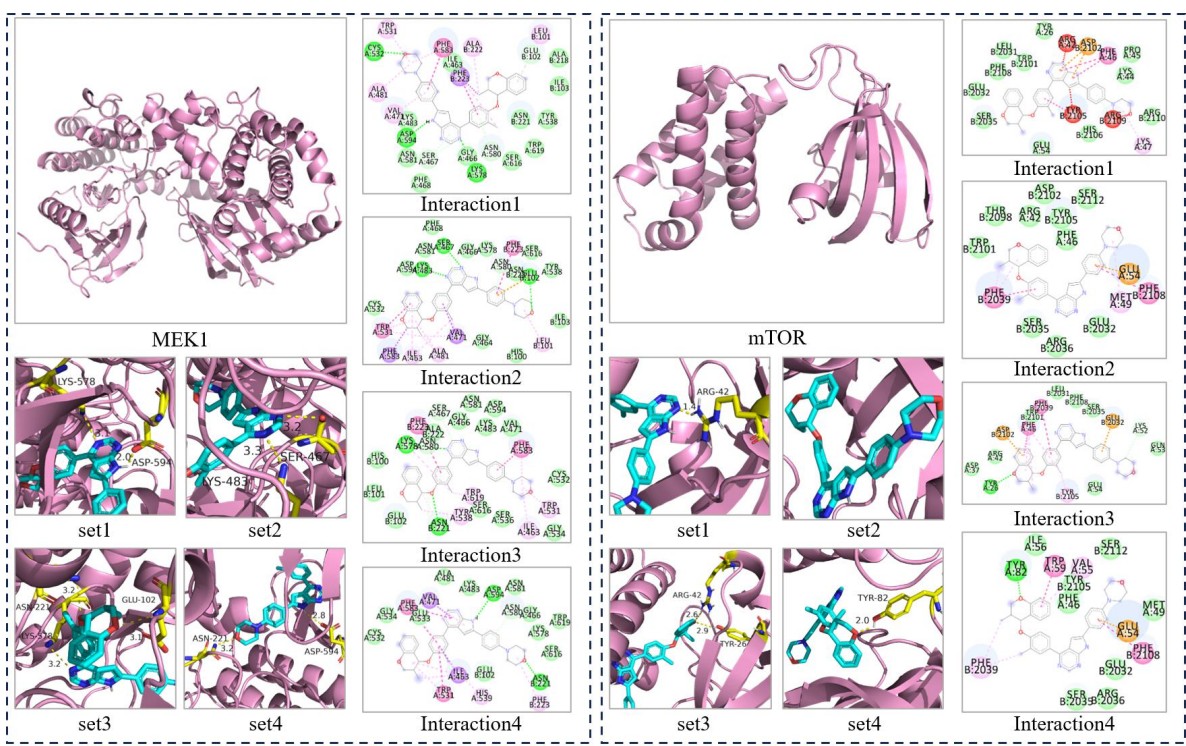

*Figure 7.* Compounds are ranked by the sum of absolute Vina scores across two targets (black). Colored markers represent individual affinities for targets such as MEK1/mTOR (left), Epro/Mpro (center), and 5HT2C/TAAR1 (right). The shaded region highlights the top 10 lead molecules.

*Figure 8.* Binding conformations of a representative multi-target molecule against tumor-related targets.

ometry with pocket-scale spatial proximity while masking ligand-irrelevant residues, enabling efficient and structurally informed generation. By jointly fusing this representation with amino acid and nucleotide sequences and conditioning generation on explicit pharmacological constraints, our method consistently produces affinity-optimized, drug-like compounds across diverse targets. These results demonstrate the effectiveness of incorporating explicit pocket geometry into generative pipelines and highlight its potential for scalable and generalizable multi-target drug design.

## Acknowlegements

The authors thank the members of Machine Learning and Artificial Intelligence Laboratory, School of Computer Science and Technology, Wuhan University of Science and Technology, for their helpful discussion within seminars. This work was supported by National Natural Science Foundation of China (61972299, 82003593) and Hubei Province Natural Science Foundation of China (No.2024AFB865).

## Code and Data Availability

The source code is publicly available at https://github.com/HaoranLiu1998/MM-MT. The datasets were obtained from the following public databases: ZINC (https://zinc.docking.org), ChEMBL (www.ebi.ac.uk/chembl), UniProt (www.uniprot.org), AlphaFold (www.alphafold.com), NCBI (www.ncbi.nlm.nih.gov), and PDB (www.rcsb.org).

## Impact Statement

This paper presents work whose goal is to advance the field of machine learning. There are many potential societal consequences of our work, none of which we feel must be specifically highlighted here.

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

## A. Pharmacological Properties

Table 3 summarizes the eight pharmacological properties used to condition molecular generation, together with their original definition domains before normalization. These properties are selected to jointly capture general drug-likeness constraints and target-specific structural requirements. The first five properties-molecular weight (MW), number of hydrogen bond donors (HBD), number of hydrogen bond acceptors (HBA), number of rotatable bonds (RotB), and the octanol-water partition coefficient (LogP)-are closely related to Lipinski's Rule of Five (RO5). Their definition ranges are chosen to cover the typical physicochemical space of orally bioavailable small molecules while allowing moderate flexibility beyond strict RO5 cutoffs to accommodate multi-target optimization. Specifically, the MW range of $[0, 500]$, HBD and HBA ranges of $[0, 5]$, RotB range of $[0, 8]$, and LogP range of $[-2, 4]$ reflect commonly accepted bounds observed in approved drug-like compounds.

The remaining three properties-number of rings (NumR), maximum ring size (MaxR), and number of amino groups (NumA)-are introduced to encode target-dependent structural preferences. These properties are particularly motivated by the characteristics of the trace amine-associated receptor 1 (TAAR1), which is a G protein-coupled receptor that requires ligand-induced activation rather than inhibition. TAAR1 possesses a compact and spatially constrained binding pocket, favoring ligands with a limited number of small rings, typically no larger than six to eight atoms. In addition, effective TAAR1 agonists often contain at least one protonatable amino group, which is critical for forming key interactions that trigger receptor activation. Accordingly, NumR, MaxR, and NumA are explicitly constrained within biologically plausible ranges to bias generation toward chemically feasible agonists. All properties are subsequently normalized to the unit interval $[0, 1]$ to enable stable and balanced conditioning during training.

*Table 3.* Eight pharmacological properties and their scaling ranges.

| Property | Full Name | Scaling Space |
|---|---|---|
| MW | Molecular Weight | $[0, 500] \rightarrow [0, 1]$ |
| HBD | H-Bond Donors | $[0, 5] \rightarrow [0, 1]$ |
| HBA | H-Bond Acceptors | $[0, 5] \rightarrow [0, 1]$ |
| RotB | Rotatable Bonds | $[0, 8] \rightarrow [0, 1]$ |
| LogP | Octanol-water partition | $[-2, 4] \rightarrow [0, 1]$ |
| NumR | Number of rings | $[0, 4] \rightarrow [0, 1]$ |
| MaxR | Max size of rings | $[3, 8] \rightarrow [0, 1]$ |
| NumA | Number of amino groups | $[0, 3] \rightarrow [0, 1]$ |

## B. Characterizing Multi-Target Ligand Size via Maximum Interatomic Distances

To quantify the spatial extent of molecules in our dataset, we compute the maximum interatomic distance for each compound, defined as the largest Euclidean distance between any two atoms in a single optimized 3D conformation. Specifically, each molecule is first embedded into three-dimensional space and locally optimized using a classical force field, after which all pairwise interatomic distances are evaluated and the maximum value is retained.

This definition provides a conservative estimate of molecular size. In practice, many drug-like compounds contain multiple rotatable bonds, allowing them to adopt more compact conformations under different torsional configurations. As a result, the maximum interatomic distance measured from a single optimized conformation may overestimate the effective spatial extent of the molecule during binding. Therefore, the reported values should be interpreted as an upper bound rather than an exact measure of molecular compactness.

Figure 9 shows the distribution of maximum interatomic distances across all ligands. Since ligand size directly determines the spatial range over which a binding pocket must accommodate ligand-protein interactions, this statistic provides a principled basis for estimating binding pocket extent. We observe that over 90% of molecules exhibit a maximum interatomic distance below 20.45 Å. Based on this conservative estimate, we adopt a cutoff of 20 Å when masking long-range spatial edges during structural topological construction, ensuring that relevant ligand-pocket interactions are retained while avoiding unnecessary graph complexity.

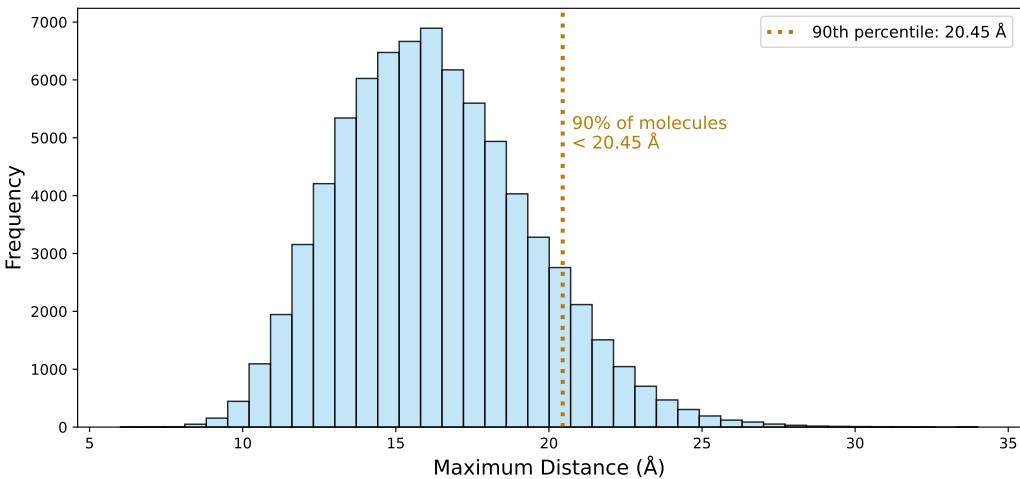

*Figure 9.* Distribution of Maximum Interatomic Distances.

## C. Dataset Construction and Preprocessing Details

To support multimodal, multi-target drug design, we construct a comprehensive dataset by integrating chemical, protein, structural, and genomic resources.

- **ZINC**: Used to pretrain the pharmacological molecule generator. We apply the Rule of Three (RO3) to filter compounds, retaining **87,719,678** drug-like molecules with molecular weight $\leq 300$ and LogP $\leq 3$.

- **ChEMBL**: From 551,223 curated ligand-target interaction pairs, we select ligands that interact with *two distinct protein targets*, yielding **90,432** valid (ligand, target$_1$, target$_2$) triplets.

- **UniProt**: Provides amino acid sequences and UniProt identifiers for all protein targets.

- **AlphaFold**: Supplies predicted three-dimensional protein structures corresponding to UniProt entries.

- **NCBI**: Nucleotide sequences are retrieved by mapping UniProt IDs to RefSeq IDs using UniProt cross-references, followed by gene-level queries to the NCBI database.

**Preprocessing steps include:**

- **SMILES standardization**: Ligands are standardized using RDKit. Molecules with token lengths outside the range [34, 74] are discarded.

- **Target filtering**: Protein targets with amino acid sequences longer than 1,000 residues or nucleotide sequences exceeding 6,400 bases are removed.

- **Pharmacological property extraction**: An 8-dimensional pharmacological property vector $\mathbf{\Phi}$ is computed for each ligand using RDKit.

- **Homology filtering**: To ensure target-level generalization, training samples whose protein targets share more than 40% sequence identity with any test protein are excluded.

After all filtering steps, the final dataset contains **75,921** high-quality multimodal multi-target samples. Each sample is represented as: $(\mathbf{c}, \mathbf{\Phi}, \{\mathcal{G}_k, \mathbf{S}_{aa,k}, \mathbf{S}_{nt,k}\}_{k=1}^2)$,

*Table 4.* Details of diseases and targets.

| Disease | Target protein 1 | | | | Target protein 2 | | | |
|---|---|---|---|---|---|---|---|---|
| | Name | Uniprot ID | PDB ID | Gene ID | Name | Uniprot ID | PDB ID | Gene ID |
| Tumour | MEK1 | P15056 | 7M0Y | NM_004333.4 | mTOR | P62942 | 1FAP | NM_054014 |
| COVID-19 | Mpro | P0DTD1 | 7ZB7 | None | Epro | P0DTC4 | 7K3G | NC_045512.2 |
| Schizophrenia | TAAR1 | P04899 | 8WCC | NM_138327.4 | 5HT2C | P28335 | 6BQG | NM_000868.4 |

## D. Details of Diseases and Targets

Table 4 provides a detailed summary of the disease settings and their associated target proteins used for evaluation. For each disease, two representative targets are selected to construct a multi-target generation scenario, covering diverse therapeutic contexts and binding mechanisms. Each target is annotated with its protein name, UniProt identifier, experimentally resolved PDB structure, and corresponding gene identifier when available, ensuring clear traceability across sequence, structure, and genomic modalities.

The tumour setting includes MEK1 and mTOR, two key kinases involved in oncogenic signaling pathways, representing targets with well-characterized allosteric and catalytic binding sites. The COVID-19 setting focuses on viral proteins Mpro and Epro, which are essential for viral replication and assembly. These targets originate from the SARS-CoV-2 genome and therefore lack conventional host gene identifiers. The schizophrenia setting comprises TAAR1 and 5HT2C, two G protein-coupled receptors implicated in neuropsychiatric regulation, capturing receptor-mediated signaling mechanisms with distinct activation and binding characteristics.

All targets listed in Table 4 have experimentally resolved structures deposited in the Protein Data Bank, enabling reliable pocket identification and structure-based modeling. Together, these disease-target pairs span enzymes, viral proteins, and membrane receptors, providing a challenging and diverse benchmark for evaluating multi-target, pharmacology-conditioned molecular generation.

## E. Protein Geometric-Topological Construction

Instead of modeling the entire protein structure as a full residue graph, we adopt a pocket-centric topological construction that focuses on ligand-interacting regions (Figure 10). Residues outside the binding pocket are masked, and edges are selectively defined by spatial proximity and backbone folding geometry, yielding a sparse yet informative topology.

Compared to full-structure graphs with quadratic edge complexity $O(N^2)$, the proposed pocket-centric representation substantially reduces both node and edge counts while preserving binding-relevant interactions. The reduction becomes more pronounced for large proteins: as shown in Figure 10, graphs with more than $1.2 \times 10^5$ nodes can be simplified by approximately $3 \times 10^4$ nodes, leading to significantly improved computational efficiency and scalability.

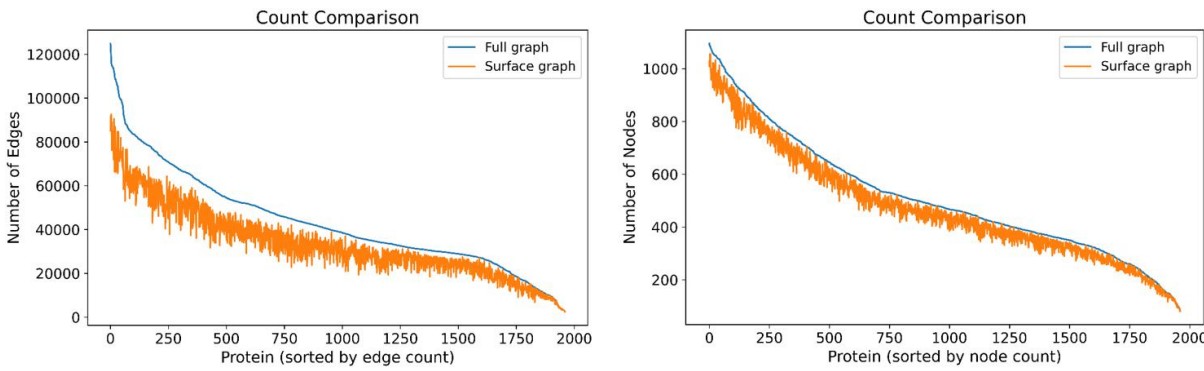

*Figure 10.* Protein topological construction preserving ligand-binding pockets while simplifying internal structure.

*Table 5.* Uniformly sampled pharmacological property ranges for molecule generation.

| MolWt | HBD | HBA | RotB | LogP | NumR | MaxR | NumA |
|-------|-----|-----|------|------|------|------|------|
| 180.0 | 1.0 | 1.0 | 1.6 | -0.8 | 0.8 | 4.0 | 0.6 |
| 260.0 | 2.0 | 2.0 | 3.2 | 0.4 | 1.6 | 5.0 | 1.2 |
| 340.0 | 3.0 | 3.0 | 4.8 | 1.6 | 2.4 | 6.0 | 1.8 |
| 420.0 | 4.0 | 4.0 | 6.4 | 2.8 | 3.2 | 7.0 | 2.4 |
| 500.0 | 5.0 | 5.0 | 8.0 | 4.0 | 4.0 | 8.0 | 3.0 |

## F. Pharmacological Property Sampling Ranges

Table 5 reports the uniformly sampled ranges of the eight pharmacological properties used to condition molecular generation and subsequent PCA analysis. Each row corresponds to a discrete sampling level, forming a structured grid over the property space to ensure balanced coverage during training and evaluation. The selected ranges span chemically reasonable intervals for drug-like molecules, covering molecular weight, hydrogen bonding capacity, flexibility, lipophilicity, and ring-related structural features.

For COVID-19 and tumour tasks, compounds are sampled uniformly across the full grid to encourage broad exploration of the pharmacological space. In contrast, for the TAAR1-related schizophrenia task, sampling is restricted to enforce a single amino group and small ring sizes, reflecting target-specific activation requirements. These controlled sampling ranges enable consistent cross-epoch comparison in PCA space while providing explicit and interpretable pharmacological constraints for molecule generation.

