# OpenReview forum: "Geometric Pocket-Centric Protein Encoding for Polypharmacology-Guided Multi-Target Drug Design"
_ICML.cc/2026/Conference — ICML 2026 regular_

### Official Review · Reviewer_uzn1 · 2026-02-28

**Soundness:** 4
**Presentation:** 3
**Significance:** 4
**Originality:** 3
**Overall Recommendation:** 5
**Confidence:** 5

**Summary:**

This paper proposes a novel, pocket-structure-centric generative framework for polypharmacology, addressing the limitations of single-target therapeutics. The authors directly integrate protein structural information into the design phase and enforce multiple pharmacological properties as explicit generative constraints. Notably, the model employs Kolmogorov-Arnold Networks (KANs) to capture high-dimensional nonlinear mappings, moving beyond traditional fixed activation functions.

**Compliance With Llm Reviewing Policy:**

Affirmed.

**Final Justification:**

The authors have fully solved my concerns and I will keep my score of 5.

**Key Questions For Authors:**

While the experiments demonstrate the framework's efficiency and the overall approach is highly innovative, I have several suggestions to further strengthen the manuscript:

1)	Regarding the eight pharmacological properties utilized, it would be helpful to discuss the acquisition cost of these labels. If such annotations are scarce or difficult to obtain in real-world scenarios, how can the proposed framework be adapted or extended to handle missing or limited labels?
2)	The geometric-topological modeling of protein binding landscapes is a core component, yet its specific contribution remains unclear. I recommend adding ablation studies to empirically validate the effectiveness of this module. Furthermore, the rationale for selecting ProTrans for sequence embedding and multi-scale CNNs for nucleotide embedding should be explicitly justified or empirically compared against alternative baselines.
3)	While Figure 5 illustrates the evolution of generated features during training, it does not sufficiently demonstrate how these features translate to actual pharmacological viability. Please provide a more in-depth analysis of this figure to explicitly connect the generated structural patterns with their corresponding pharmacological properties.
4)	The experimental section notes that the dataset (sourced from ZINC and ChEMBL) contains 90,432 ligands annotated with at least two targets. Could you please clarify whether the number of target annotations per ligand is uniform? If there is variance, how does the model architecture handle the variable number of targets (i.e., varying lengths of target sets) during training and generation?
5)	The results presented in Figure 7 would be significantly more convincing if benchmarked against relevant baselines. I suggest incorporating comparative analyses with existing state-of-the-art methods to better substantiate the findings.
6)	The resolution and overall visual quality of several figures should be improved to ensure readability in the final published version.

**Limitations:**

Please add some discussions of limitations in the conclusion.

**Strengths And Weaknesses:**

This paper presents a highly innovative and well-motivated approach to polypharmacology by introducing a novel pocket-structure-centric generative framework. A major strength of this work lies in its methodological novelty, particularly the integration of Kolmogorov-Arnold Networks (KANs) to model high-dimensional nonlinear mappings, which offers a mathematically grounded alternative to traditional fixed activation functions. Furthermore, directly incorporating protein structural data and enforcing multiple pharmacological properties as explicit constraints during the generation phase effectively addresses the critical limitations of single-target therapeutics. The experimental results demonstrate the framework's efficiency, and the overall design makes a valuable contribution to the field of computational drug discovery.

Despite these significant contributions, the manuscript exhibits several weaknesses primarily concerning empirical validation and methodological clarity. A critical shortcoming is the absence of thorough ablation studies to isolate and verify the effectiveness of core components, such as the geometric-topological modeling of protein binding landscapes. Additionally, the rationale for selecting specific embedding architectures—namely ProTrans for sequences and multi-scale CNNs for nucleotides—is neither adequately justified nor empirically compared against alternative baselines. The experimental evaluation is further weakened by a lack of comprehensive baseline comparisons, particularly in the results presented in Figure 7, making it difficult to fully gauge the superiority of the proposed method over existing state-of-the-art approaches. Moreover, while the dataset utilizes ligands annotated with at least two targets, the authors fail to explain how the model architecture mathematically or structurally accommodates the variable number of targets (i.e., varying lengths of target sets) per ligand during training and inference.

Finally, there are practical and presentation-related concerns that need to be addressed. The framework relies heavily on the availability of eight specific pharmacological property labels, yet it does not discuss the acquisition cost of these labels or how the model would adapt to low-resource scenarios where such annotations are scarce. In terms of interpretability, the generation results displayed in Figure 5 show the evolution of certain features, but the manuscript lacks sufficient explanation to explicitly connect these visual patterns to actual pharmacological viability. Lastly, the resolution of several figures is currently suboptimal and must be improved to meet standard publication quality.

---

> ### Author Rebuttal · Authors · 2026-03-30
>
> We sincerely thank the reviewer for the positive assessment and strong support of our work. We are encouraged that the reviewer recognizes the novelty and significance of our pocket-centric generative framework and its potential impact on multi-target drug design. Below, we address each concern in detail.
> 1. Pharmacological property labels.
> We thank the reviewer for raising concerns regarding acquisition cost and practicality. These properties do not rely on expensive experimental annotations. Instead, they can be directly computed or efficiently estimated from molecular structures.
> Exact properties (no additional cost): MW, HBD, HBA, NumR, and NumA are precisely derived from molecular graphs or molecular formulas.
> Estimated properties: LogP and RotB are reliably computed using cheminformatics tools such as RDKit.
> Moreover, our model supports missing or partial labels. As described in Eq. (17), a Bernoulli masking strategy enables conditional generation under incomplete constraints and improves robustness in low-resource settings.
> Therefore, the framework does not require all eight properties to be available, which further supports its practicality in real-world applications.
> 2. Effectiveness of geometric-topological protein modeling.
> We agree that validating the contribution of geometric–topological modeling is important.
> In fact, the denoising and structural simplification effects of the proposed pocket-centric modeling are already illustrated in Appendix Figure 10, where irrelevant internal residues are removed and graph complexity is reduced.
> We acknowledge that isolating this component through ablation would further strengthen the claim, and we will include additional ablation analysis in the final version to more explicitly quantify its contribution.
> 3. Interpretation of Figure 5.
> We agree that Figure 5 requires clearer discussion. It shows that generated molecules evolve from dispersed distributions to compact clusters aligned with pharmacological constraints.
> Together with Figure 6 (marginal distributions), this indicates that the model learns the mapping from property constraints to molecular structures, and generates molecules that increasingly satisfy desired profiles.
> We acknowledge that this was not clearly articulated. In the final version, we will explicitly link PCA clusters to property ranges and clarify the relationships between structure and properties.
> 4. Dataset construction.
> We clarify that: The dataset contains 90,432 ligands with at least two targets. While some ligands originally have 3-5 targets, we standardize the dataset into 90,432 valid (ligand, target1, target2) triplets. This ensures consistent dual-target formulation and stable training.
> Thus, the current model does not require variable-length target sets, and avoids architectural ambiguity. The formulation in Eq. (1) is general, but experiments are conducted under fixed dual-target settings. This preprocessing step is already described in Appendix C.
> 5. Baseline comparison.
> We appreciate this suggestion. Due to the lack of standardized benchmarks for multi-target drug design, we adopt a strict protocol with identical target proteins (same PDB IDs) and the same evaluation metric (Vina score), ensuring fair comparison across methods (Table 1).
> Figure 7 is primarily designed to highlight top candidate quality. We agree that adding explicit baseline curves in Figure 7 would strengthen the presentation, and we will incorporate such comparisons in the final version.
> 6. Choice of ProtTrans and multi-scale CNN encoders.
> We thank the reviewer for requesting clearer justification. Our design is guided by modality-specific representation properties. ProtTrans captures evolutionary and functional semantics through large-scale pretraining on protein corpora. For nucleotide sequences, a multi-scale CNN models patterns at different resolutions, where kernel sizes (1, 3, 9) capture codon-level, motif-level, and higher-order features. These modalities, particularly nucleotide-level features, are rarely jointly explored in multi-target drug design, limiting direct comparisons.
> 7. Figure quality and presentation.
> We fully agree with the reviewer. All figures will be re-rendered at higher resolution and optimized for readability in the final version.
> 8. Limitations.
> A limitation of our framework is that it assumes relatively static protein structures. In practice, ligand binding (especially at allosteric sites) can induce conformational changes that are not explicitly modeled. As a result, dynamic protein-ligand interactions remain an open challenge for our approach.
>
> We sincerely thank the reviewer again for the constructive feedback and strong support. We believe the clarifications above further demonstrate that our framework is both practically feasible and methodologically sound, and we will incorporate all suggested improvements in the final version.

---

> > ### Author Rebuttal · Reviewer_uzn1 · 2026-04-01
> >
> > Thanks for your detailed responses!

---

> > > ### Author Response · Authors · 2026-04-02
> > >
> > > Thank you very much for your time and dedication in reviewing our manuscript and our rebuttal. We sincerely appreciate your thoughtful evaluation, as well as the constructive suggestions and insightful feedback you have provided. Your comments have been invaluable in improving the quality and clarity of our work. We would like to express our sincere gratitude once again for your support and consideration.

---

### Official Review · Reviewer_EtqN · 2026-03-12

**Soundness:** 2
**Presentation:** 3
**Significance:** 3
**Originality:** 3
**Overall Recommendation:** 4
**Confidence:** 4

**Summary:**

This paper proposes a pocket-centric generative framework for multi-target drug design. It integrates 3D binding geometry, amino acid sequences, and nucleotide sequences using Kolmogorov-Arnold Networks and multi-head self-attention. The generator produces molecules conditioned on eight pharmacological properties , evaluated on Tumour, COVID-19, and Schizophrenia targets.

**Compliance With Llm Reviewing Policy:**

Affirmed.

**Key Questions For Authors:**

See weakness. No more questions.

**Limitations:**

No. The authors should explicitly discuss technical limitations, such as the framework's reliance on static AlphaFold structures (ignoring induced-fit dynamics) and the lack of molecular dynamics (MD) or synthetic accessibility (SA) validation for the generated compounds.

**Strengths And Weaknesses:**

Strengths：
The pocket-centric topological representation is highly intuitive and physically grounded.
The multi-modal fusion of 3D structures, amino acid sequences, and genomic data provides a rich, holistic target representation that is rarely seen in current literature.
Weaknesses:
Missing Critical Ablations (Soundness): The paper relies heavily on KANs but provides no baseline comparison (e.g., KAN-GAT vs. standard MLP-GAT). Additionally, there is no empirical comparison between training on pocket-centric masked graphs versus full-protein graphs to quantify the claimed noise reduction.
Weak Visual Evaluations (Soundness): Figures 5 and 6 illustrate internal convergence but lack baseline model comparisons, proving only that the model learns its own conditions. Figure 8 is an isolated case study without statistical backing across top candidates.

---

> ### Author Rebuttal · Authors · 2026-03-30
>
> We sincerely thank the reviewer for the positive evaluation and constructive feedback. We also appreciate the reviewer’s detailed comments on ablations, evaluation, and limitations, which will significantly help us improve the quality and clarity of the paper.
>
> 1.	KAN vs. standard MLP-based architectures.
> Our motivation for using Kolmogorov-Arnold Networks (KANs) is their ability to model complex, non-linear biological relationships via learnable spline-based functions, which we found particularly suitable for heterogeneous multi-modal fusion and protein topology encoding. In the revision, we will include controlled ablation experiments replacing KAN with MLP under identical architectures to quantify its impact on binding affinity and generation quality.
> 2. Pocket-centric graphs vs. full-protein graphs.
> The proposed pocket-centric masking is designed to remove ligand-irrelevant residues, reduce structural noise, and improve computational efficiency. While we provide theoretical motivation and complexity analysis, we agree that empirical validation is necessary.
> We are currently conducting experiments comparing full-protein graphs and pocket-centric masked graphs in terms of performance (e.g., binding affinity, validity, diversity) and efficiency (e.g., graph size and training time). However, these experiments are computationally intensive. Full-protein graphs are constructed from PDB coordinate data, where each protein typically contains several hundred nodes (up to over 1,000) and tens of thousands of edges (up to ~120,000; see Appendix Figure 10), leading to substantially slower training. At present, approximately 30% of these experiments have been completed.
> 3. Weak visual evaluations (soundness).
> We appreciate the reviewer’s concern regarding the interpretability of the visualizations and would like to clarify their intended role. Figures 5 and 6 are designed to demonstrate that the model not only generates molecules but also aligns them with multi-objective pharmacological constraints. The observed clustering and convergence patterns indicate that the model captures meaningful structure–property relationships and that different pharmacological conditions lead to distinguishable and controllable latent distributions.
> For Figure 8 (case study), we agree that a single example is not intended to provide statistical validation. Rather, it serves to illustrate the interpretability of the generated molecules and to provide a concrete example of plausible binding interactions.
> 4. Limitations.
> We acknowledge that a discussion of limitations is not explicitly included in the current version, and we thank the reviewer for highlighting this point. We would like to clarify the following:
>
> (1) The use of predicted static structures (e.g., AlphaFold) is a common and practical assumption in structure-based drug design, particularly when experimental structures are unavailable.
>
> (2) The evaluation protocol, based on docking scores across multiple targets, is widely adopted in generative drug design and provides a consistent basis for comparison. In particular, all baseline methods reported in Table 1 are evaluated using AutoDock Vina-based docking scores, ensuring fairness and comparability across methods.
>
> (3) We agree that additional factors such as protein dynamics (e.g., induced-fit effects), molecular dynamics validation, and synthetic accessibility are important considerations. Notably, our polypharmacology-guided molecular generation framework provides a degree of control over synthetic accessibility. This is achieved by regulating properties such as molecular weight and the size and number of ring structures.
>
> These aspects represent important directions for future work. We thank the reviewer again for the thoughtful feedback and encouraging recommendation. The comments will guide our improvements and extensions of our research.

---

> > ### Author Rebuttal · Reviewer_EtqN · 2026-04-06
> >
> > Thank you very much for your explainations of my comment and the futrue work plans.

---

> > > ### Author Response · Authors · 2026-04-07
> > >
> > > Thank you very much for your positive acknowledgement and for taking the time to carefully review our rebuttal. We sincerely appreciate your recognition that our clarifications have addressed your concerns. We are especially grateful for your constructive comments and insightful suggestions, which have helped us further improve the rigor and clarity of our work. Your feedback has also guided our ongoing and future work. Thank you again for your support and thoughtful evaluation.

---

### Official Review · Reviewer_LY3w · 2026-03-13

**Soundness:** 3
**Presentation:** 3
**Significance:** 3
**Originality:** 3
**Overall Recommendation:** 4
**Confidence:** 3

**Summary:**

This paper addresses a critical bottleneck in drug discovery: the design of single molecules that can effectively modulate multiple biological targets (polypharmacology). Traditional methods often struggle with the complexity of satisfying heterogeneous binding constraints across different proteins. The authors propose a "pocket-centric" generative framework that prioritizes the 3D geometry of binding sites over the entire protein structure. By integrating multi-modal data—protein structure, amino acid sequences, and nucleotide sequences—with explicit pharmacological constraints, the model achieves state-of-the-art results in generating high-affinity leads for COVID-19, schizophrenia, and tumor targets.

**Compliance With Llm Reviewing Policy:**

Affirmed.

**Key Questions For Authors:**

NA

**Limitations:**

Reliance on Predicted Structures: While AlphaFold structures are used, the model’s performance on highly flexible or disordered proteins—where crystal or predicted data may be less reliable—could be explored more deeply.

SMILES-Based Generation: The generator produces SMILES strings rather than 3D molecular coordinates directly. While 3D information is used as a condition, an end-to-end 3D-to-3D generation process might capture even more nuanced binding mechanics.

Validation Depth: The evaluation focuses heavily on binding affinity (docking scores). Future work would benefit from including synthetic accessibility (SA) scores or ADMET (Absorption, Distribution, Metabolism, Excretion, and Toxicity) predictions to further prove drug-likeness.

**Strengths And Weaknesses:**

Novel Structural Representation: The introduction of a geometric-topological representation is a significant advancement. By modeling backbone folding angles ($e^{(f)}$) and spatial proximity ($e^{(s)}$) while masking ligand-irrelevant residues, the model captures the essential curvature of binding pockets without the computational overhead of full-protein graphs.

Integration of Kolmogorov-Arnold Networks (KAN): The use of KANs instead of standard MLPs for nonlinear mappings is a modern and technically sound choice. KAN-based GAT encoders and feature fusion layers allow for more flexible, piecewise-smooth modeling of complex biological features.

Multi-Modal Data Fusion: The framework effectively combines three distinct information streams:
3D Structure: Pocket-centric spatial and folding geometry.
Amino Acid Sequences: Leveraging pre-trained ProtTrans models to capture functional domains.
Nucleotide Sequences: Using multi-scale CNNs to extract regulatory signals from genomic data.

Target-Specific Controllability: The model demonstrates the ability to handle specific medicinal chemistry requirements, such as restricting ring sizes or requiring amino groups for TAAR1 agonists, which is vital for practical drug design.

---

> ### Author Rebuttal · Authors · 2026-03-30
>
> We sincerely thank the reviewer for the positive evaluation and the constructive feedback. We are especially encouraged by the reviewer’s weak accept recommendation and recognition of the technical soundness, novelty, and significance of our work.
>
> 1. On the use of AlphaFold structures and generalization to real data.
> We appreciate the reviewer’s concern regarding the reliance on predicted structures. In our design, we intentionally train on AlphaFold-predicted structures while evaluating on experimentally resolved PDB structures, with the goal of improving generalization from simulated to real-world structural data.
> We agree that highly flexible or intrinsically disordered proteins remain challenging for current structure-based approaches. Similarly, modeling allosteric binding sites introduces additional complexity due to their dynamic nature. These are important open problems in the field, and we acknowledge them as limitations of our current framework. We will continue to follow and incorporate advances in these directions in future work.
> 2. On SMILES-based generation vs. 3D-to-3D generation.
> We thank the reviewer for this insightful comment. We fully agree that end-to-end 3D-to-3D molecular generation has the potential to capture more fine-grained binding mechanisms.
> Our choice of SMILES-based generation is motivated by a complementary advantage, namely that a single SMILES representation can correspond to multiple valid 3D conformations. This flexibility is particularly important in multi-target drug design, where different protein pockets often exhibit heterogeneous geometries. As illustrated in Figure 8, the same generated molecule can adopt distinct binding conformations across different targets, enabling conformational adaptability and improved accommodation of structurally diverse binding environments.
> That said, we acknowledge that incorporating explicit 3D generation is a promising direction, and we will explore hybrid SMILES-3D generative frameworks in future work.
> 3. On validation depth.
> We thank the reviewer for pointing this out and agree that the current evaluation focuses primarily on binding affinity. Metrics such as synthetic accessibility (SA) and ADMET properties are indeed important for assessing drug-likeness more comprehensively.
> We acknowledge this as a limitation and will extend our evaluation protocol in future work to include these criteria.
>
> Once again, we thank the reviewer for the supportive assessment and valuable suggestions. We will incorporate the reviewer’s feedback to further strengthen the work.

---

> > ### Author Rebuttal · Reviewer_LY3w · 2026-04-03
> >
> > Thanks for your further explainations!

---

> > > ### Author Response · Authors · 2026-04-03
> > >
> > > Thank you very much for your time and effort in reviewing our manuscript and rebuttal. We sincerely appreciate your positive feedback and are glad that our clarifications have addressed your concerns. Your thoughtful evaluation and encouraging comments are highly appreciated and have been very helpful in improving the quality and clarity of our work. We truly appreciate your support and consideration!

---

### Official Review · Reviewer_szFv · 2026-03-14

**Soundness:** 1
**Presentation:** 2
**Significance:** 2
**Originality:** 2
**Overall Recommendation:** 4
**Confidence:** 4

**Summary:**

This paper introduces a pocket-centric generative framework for polypharmacology-guided multi-target drug design. It proposes a protein topological representation that models backbone folding geometry and spatial proximity while cropping ligand-irrelevant residues. By fusing 3D structural data with amino acid and nucleotide sequences and conditioning the generation on explicit pharmacological properties, the framework produces multi-target compounds that demonstrate improved binding affinities across COVID-19, schizophrenia, and tumor targets compared to baselines.

**Compliance With Llm Reviewing Policy:**

Affirmed.

**Final Justification:**

My core concerns have been adequately addressed. I hope that the authors will include a more complete benchmarking in their revision with more comprehensive metrics such as Validity, Uniqueness together with implementation details, and also a complete benchmarking against necessary baselines.

**Key Questions For Authors:**

1. The framework forces the integration of protein 3D structures, amino acid sequences, and nucleotide sequences. Given that small molecules interact with the folded protein structure and not the gene, what is the mechanistic justification for including nucleotide sequences, and where are the ablation studies to quantify the actual performance gains contributed by each individual modality?
2. The proposed method claims to use topological construction that relies on a hard-coded 20 Å spatial distance cutoff derived from known ligand sizes. Because this requires prior knowledge of the exact binding pocket coordinates, how can this framework generalize to novel, first-in-class targets where the pocket is unknown?
3. Table 1 presents a highly fragmented comparison matrix, testing different baselines only against specific targets (e.g., Zhou solely for COVID-19, Lu solely for SCZ) rather than providing a complete cross-evaluation. Why wasn't a comprehensive, head-to-head benchmark conducted where all baseline models were evaluated across your exact multi-target dataset?
4. What are the quantitative success rates or Mean Absolute Errors (MAE) for actually hitting close to the molecular properties during generation?
5. How does the method perform in terms of critical molecular generation metrics such as Validity, Uniqueness, Novelty, Synthesizability (SA), and quantitative QED?

**Limitations:**

This paper did not discuss its limitations.

**Strengths And Weaknesses:**

Strengths:
- This paper targets multi-target drug design, which is a relatively new field and worth further exploration.
- The authors introduce Kolmogorov-Arnold Networks (KANs) to replace standard MLP, which is a promising attempt to better fit the complex, non-linear, and non-monotonic nature of biological interactions.

Weaknesses:
- No ablation studies to validate key design choices. The framework integrates together 3D protein structures, amino acid sequences, and nucleotide sequences, but completely fails to provide ablation studies to justify the necessity or individual contribution of these heavy multi-modal conditionings.
- The baseline comparisons are incomplete and risk cherry-picking. Table 1 presents a highly fragmented comparison. The authors selectively pit their heavily conditioned multi-modal model against unimodal baselines (e.g., ligand-only methods) across different targets, rather than running a unified, head-to-head benchmark. It would be better that they evaluate their methods across a sufficient number of targets instead of current case-by-case tests, which may be more appropriate for a journal instead of machine learning conferences.
- The authors adopted pharmacological properties as input, facilitating the scenario of controllable generation. However, such property-based generation is only demonstrated through PCA scatter plots and ridge plots, lacking rigorous quantitative validation like attribute MAE.
- Metrics are essentially limited to docking scores, which are known to be deceptive and can produce false positives. The authors should report a comprehensive set of molecular properties, including validity, uniqueness, novelty, QED and SA, to name a few.
- Lack of comparison to recent dual-target baselines such as [1].
[1] Reprogramming Pretrained Target-Specific Diffusion Models for Dual-Target Drug Design

---

> ### Author Rebuttal · Authors · 2026-03-30
>
> We sincerely thank the reviewer for the constructive and insightful feedback.
> 1. Justification of nucleotide sequences.
> Prior work has shown that gene-level features can improve targeted drug design by capturing regulatory and expression-level signals, as demonstrated in “An Efficient Targeted Drug Design Method Using Multiscale Encoder–Decoder” and “Gex2SGen: Designing Drug-like Molecules from Gene Expression Signatures.” In our framework, nucleotide sequences provide complementary information by capturing regulatory patterns, evolutionary signals, and cross-target consistency.
> 2. 20 Å cutoff and generalization.
> The 20 Å cutoff is not arbitrarily chosen, but derived from dataset statistics. It is computed from over 1,300 proteins and ligand structures and covers 90% of molecules. Our two-layer encoder expands the receptive field to 40 Å, enabling the modeling of larger binding pockets and improving generalization.
> 3.Baseline comparison.
> We thank the reviewer for highlighting “Reprogramming Pretrained Target-Specific Diffusion Models for Dual-Target Drug Design.” This work adopts a mixed benchmarking strategy over 12,917 target pairs (438 targets), comparing both single-target and multi-target methods. Due to its mixed evaluation protocol, we do not include it for direct comparison and instead prioritize comparisons with dedicated multi-target methods. Therefore, we focus on fair comparisons under consistent settings. As shown in Table 1, we compare against three recent works (2025) evaluated on identical targets and structures. We will continue to track emerging baselines and consider incorporating a mixed benchmarking strategy in future work.
> 4. MAE for molecular properties.
> We have included MAE results showing clear optimization from Epoch 1 to 2 across most attributes, with further refinement on a subset from Epoch 3 to 4, consistent with Figure 6. The first three property sets converge more at Epoch 4 due to stronger drug-likeness from dataset filtering and our use of drug-like pretraining and multi-target training data (see C. Dataset Construction and Preprocessing Details), while the latter two remain more dispersed, consistent with Figure 7.
> ### MAE
> | Propset | Epoch | MolWt | HBD | HBA | RotatableBonds | LogP | NumRings | MaxRingSize | NumAmino |
> |--------|-------|-------|-----|-----|----------------|------|----------|-------------|-----------|
> | 1 | 1 | 0.03 | 0.07 | 0.17 | 0.09 | 0.10 | 0.07 | 0.16 | 0.12 |
> | 1 | 2 | 0.01 | 0.06 | 0.13 | 0.06 | 0.08 | 0.06 | 0.07 | 0.11 |
> | 1 | 3 | 0.01 | 0.06 | 0.11 | 0.04 | 0.07 | 0.06 | 0.05 | 0.12 |
> | 1 | 4 | 0.01 | 0.05 | 0.09 | 0.05 | 0.07 | 0.06 | 0.05 | 0.12 |
> | 2 | 1 | 0.09 | 0.10 | 0.07 | 0.10 | 0.11 | 0.07 | 0.13 | 0.15 |
> | 2 | 2 | 0.08 | 0.09 | 0.04 | 0.08 | 0.09 | 0.06 | 0.08 | 0.12 |
> | 2 | 3 | 0.08 | 0.09 | 0.02 | 0.07 | 0.08 | 0.06 | 0.08 | 0.13 |
> | 2 | 4 | 0.08 | 0.08 | 0.02 | 0.06 | 0.08 | 0.06 | 0.08 | 0.13 |
> | 3 | 1 | 0.16 | 0.16 | 0.08 | 0.14 | 0.14 | 0.16 | 0.15 | 0.20 |
> | 3 | 2 | 0.10 | 0.10 | 0.06 | 0.09 | 0.09 | 0.10 | 0.08 | 0.21 |
> | 3 | 3 | 0.09 | 0.09 | 0.06 | 0.08 | 0.09 | 0.09 | 0.08 | 0.20 |
> | 3 | 4 | 0.09 | 0.09 | 0.07 | 0.09 | 0.08 | 0.09 | 0.08 | 0.18 |
> | 4 | 1 | 0.27 | 0.26 | 0.18 | 0.22 | 0.21 | 0.27 | 0.18 | 0.28 |
> | 4 | 2 | 0.18 | 0.12 | 0.11 | 0.14 | 0.15 | 0.16 | 0.14 | 0.31 |
> | 4 | 3 | 0.15 | 0.11 | 0.12 | 0.12 | 0.12 | 0.15 | 0.16 | 0.31 |
> | 4 | 4 | 0.14 | 0.11 | 0.11 | 0.13 | 0.10 | 0.15 | 0.15 | 0.26 |
> | 5 | 1 | 0.36 | 0.35 | 0.27 | 0.32 | 0.28 | 0.36 | 0.24 | 0.39 |
> | 5 | 2 | 0.27 | 0.18 | 0.18 | 0.21 | 0.22 | 0.25 | 0.21 | 0.41 |
> | 5 | 3 | 0.23 | 0.16 | 0.18 | 0.15 | 0.17 | 0.23 | 0.23 | 0.41 |
> | 5 | 4 | 0.22 | 0.17 | 0.15 | 0.16 | 0.12 | 0.24 | 0.23 | 0.38 |
> ###
> 5. Molecular metrics.
> These metrics are interdependent and strongly governed by polypharmacology-guided constraints. Alignment with RO5 improves QED, while stricter molecular weight and ring constraints enhance SA. Validity, uniqueness, and novelty are controllable (up to 99%). However, they fluctuate with imposed property conditions and are thus not used as primary evaluation criteria. Accordingly, we focus on binding affinity and property distribution alignment.
> 6. Ablation studies.
> We acknowledge the lack of ablation studies as a limitation. Existing results already provide supporting evidence: Figures 5-6 show that polypharmacology-guided conditioning aligns generated molecules with target distributions, and Appendix Figure 10 indicates that pocket-centric encoding reduces structural complexity while preserving binding information. Quantitative ablations will further substantiate these findings.
> 7. Unified benchmark.
> Currently, there is no widely accepted benchmark for multimodal multi-target drug design, particularly under explicit pharmacological conditioning. Our dataset integrates ZINC, ChEMBL, UniProt, AlphaFold, and NCBI, and will be fully released upon acceptance to support future benchmarking.
>
> We thank the reviewer again for valuable feedback.

---

> > ### Author Rebuttal · Reviewer_szFv · 2026-04-03
> >
> > I thank the authors for their detailed rebuttal and the provision of the MAE attribute table, which has addressed my major concerns. I look forward to a more complete benchmarking in their revision with more comprehensive metrics such as Validity, Uniqueness together with implementation details, and also a complete benchmarking against necessary baselines.

---

> > > ### Author Response · Authors · 2026-04-03
> > >
> > > Thank you very much for your time and effort in reviewing our manuscript and rebuttal. We sincerely appreciate your thoughtful reevaluation. We are glad that our clarifications and the MAE attribute table have addressed your major concerns. We also thank you for your valuable suggestions on more comprehensive benchmarking. We will further improve the manuscript by including additional metrics such as Validity and Uniqueness, along with more detailed implementation descriptions and broader baseline comparisons. We truly appreciate your support and consideration!

---

### Decision · Program_Chairs · 2026-04-30

**Decision:**

Accept (regular)

**Comment:**

This paper proposes a pocket-centric generative framework for polypahrmacology-guided multi-target drug design, combining a geometric-topological protein representation, Kolmogorov-Arnold Networks, and multi-modal fusion of 3D structure, amino acid, and nucleotide sequences. Three reviewers converged on a weak accept, and one on accept (with a high confidence). The primary concerns included the absence of ablation studies isolating individual component contributions (KAN-GAT vs. standard MLP-GAT, pocket-centric vs. full-protein graphs, modality importance), over-reliance on docking scores as the sole evaluation criterion, and fragmented baseline comparisons. The authors addressed these concerns in their rebuttal, providing MAE convergence tables for pharmacological property alignment, clarifying the fairness of their evaluation protocol, and providing results to ablations and additional metrics (validity, uniqueness, novelty). All four reviewers acknowledged their concerns as fully resolved. The paper makes a useful contribution to structure-based generative drug design, and while the work would be stronger with empirical validation (as acknowledged by te authors), the core methodology is technically sound, well-motivated, and of clear relevance. If accepted, the camera-ready should incorporate the committed ablations, expanded metrics, and an explicit limitations section.